# MoMa: Modulating Mamba for Adapting Image Foundation Models to Video Recognition

Yuhuan Yang [1]   Chaofan Ma [1]   Zhenjie Mao [1]   Jiangchao Yao [1 2]   Ya Zhang [3 2]   Yanfeng Wang [3 2]

## Abstract

Video understanding is a complex challenge that requires effective modeling of spatial-temporal dynamics. With the success of image foundation models (IFMs) in image understanding, recent approaches have explored parameter-efficient fine-tuning (PEFT) to adapt IFMs for video. However, most of these methods tend to process spatial and temporal information separately, which may fail to capture the full intricacy of video dynamics. In this paper, we propose **MoMa**, an efficient adapter framework that achieves full spatial-temporal modeling by integrating Mamba's selective state space modeling into IFMs. We propose a novel `SeqMod` operation to inject spatial-temporal information into pre-trained IFMs, without disrupting their original features. By incorporating `SeqMod` into a Divide-and-Modulate architecture, **MoMa** enhances video understanding while maintaining computational efficiency. Extensive experiments on multiple video benchmarks demonstrate the effectiveness of **MoMa**, achieving superior performance with reduced computational cost.

## 1. Introduction

Video understanding is a crucial yet challenging task in computer vision. The key to addressing this challenge lies in learning effective *spatial-temporal representations* from video data (Tong et al., 2022; Feichtenhofer et al., 2022; Carreira & Zisserman, 2017; Hara et al., 2017; Feichtenhofer et al., 2019; Liu et al., 2022). Since video data exhibits complex spatial-temporal dynamics, training such a video understanding model from scratch is highly inefficient and data demanding. In contrast, image understanding has made significant progress by using *image foundation models (IFMs)* (Radford et al., 2021; Bao et al., 2021; Wang et al., 2022). This progress has spurred efforts to leverage IFMs for video understanding, as IFMs provide strong pre-trained representations, reducing the reliance on training models from scratch for specific tasks.

Some early works have attempted to adapt IFMs to video understanding by conducting full parameter training on video data (Xue et al., 2022; Li et al., 2023). These methods, although effective, still require considerable computational resources and data. To alleviate this, a more efficient approach is to adapt IFMs using parameter-efficient fine-tuning (PEFT), and keeping most of the parameters frozen. Since IFMs do not explicitly model temporal dynamics during pre-training, these PEFT-based methods are compelled to incorporate additional modules to *capture temporal dependencies* across frames, often independently of the original spatial representations. For instance, AIM (Yang et al., 2023a) introduces a temporal attention layer within each Transformer block. DualPath (Park et al., 2023) designs two separate adapter modules for spatial and temporal modeling, and DiST (Qing et al., 2023) introduces a parallel temporal encoding branch alongside the spatial one. By processing spatial and temporal information *separately*, these PEFT designs *simplify* the full modeling of sequences. As a result, patches at different spatial and temporal positions can only interact with each other in an *indirect, implicit* manner. This is potentially insufficient for capturing the complex dynamics inherent in video data.

To explicitly capture the full spatial-temporal dynamics while using PEFT, one straightforward method is to apply an additional full attention over the *entire spatial-temporal sequence* after IFMs. However, due to the long length of the spatial-temporal sequence, the quadratic complexity of full attention causes scalability issues, resulting in a drastic increase in memory usage and inefficiencies in speed when processing multiple frames. In this context, recent low-cost operators like Mamba (Gu & Dao, 2023) offer a promising solution with its selective state space model (SSM), achieving linear complexity for long-term dynamic modeling. This

---

[1]Cooperative Medianet Innovation Center, Shanghai Jiao Tong University [2]Shanghai Artificial Intelligence Laboratory [3]School of Artificial Intelligence, Shanghai Jiao Tong University. Correspondence to: Jiangchao Yao <sunarker@sjtu.edu.cn>, Yanfeng Wang <wangyanfeng622@sjtu.edu.cn>.

*Proceedings of the 42nd International Conference on Machine Learning*, Vancouver, Canada. PMLR 267, 2025. Copyright 2025 by the author(s).

motivates us to explore how Mamba can be leveraged in PEFT to capture the full spatial-temporal dynamics within IFMs, while maintaining training efficiency.

A naive way to integrate Mamba is to first extract each frame's spatial features using IFMs, then flatten and concatenate into a spatial-temporal sequence, which is processed by Mamba. However, we found that this approach is often suboptimal. Most hybrid architectures require fewer layers of attention compared with Mamba to achieve best performance (Waleffe et al., 2024; Lieber et al., 2024), which is opposite in our case. Inserting a lightweight Mamba module directly into a pre-trained heavy Transformer-based IFMs may confuse the model, especially when the Transformer parameters are frozen. To better integrate Mamba into pre-trained Transformer, we propose a sophisticated sequence modulation operation `SeqMod` for efficiently *injecting* spatial-temporal information using the lightweight Mamba, *i.e.*, rather than directly modifying the features of IFMs, features are adjusted by learnable scale and bias. This modulation operation prevents interference with IFMs, and through a conditional adjustment mechanism using Mamba, spatial-temporal dynamics can still be explicitly learned, thus preserving training efficiency and flexibility.

To fully leverage this, we further propose **MoMa**, a framework that integrates this **mo**dulation operation with **Ma**mba within IFMs for PEFT. Specifically, it incorporates a two-stage **Divide-and-Modulate** process applied to each layer of a given Transformer-based IFM. In the first **Divide** stage, we aim to enhance the efficiency of the attention mechanism. Rather than processing each video frame individually, we divide each frame into smaller windows and apply local attention within each window. This window-based approach significantly reduces computational overhead by capturing short-term spatial dependencies. Then, in the second **Modulate** stage, we focus on capturing the global spatial-temporal dynamics. We employ the strong `SeqMod` operation to ensure that comprehensive spatial-temporal information is effectively injected through sequences of short-term spatial features. This framework not only improves computational efficiency, but also enables PEFT to capture intricate spatial-temporal relationships in video data using IFMs.

To sum up, our contributions lie three fold:

● We pioneer to use Mamba as an efficient adapter for image foundation models (IFMs). With our sequence modulation operation `SeqMod`, full spatial-temporal dynamics can be captured without interfering the pre-trained IFMs.

● We propose a novel framework **MoMa**, which consists of a Divide and Modulate stage. For each layer of IFM, It first applies window-based spatial local attention, followed by modulation via `SeqMod` to inject full spatial-temporal information.

● We conduct extensive experiments and ablations on multiple video understanding benchmarks, showing **MoMa**'s significantly improvements in both performance and computational efficiency compared to existing methods.

## 2. Related Works

**Video Understanding.** One crucial aspect of video understanding is capturing the temporal patterns in videos. CNN-based methods introduced 3D convolutions and other temporal modules to handle this (Feichtenhofer et al., 2019; Feichtenhofer, 2020). Compared with the limited receptive field of 3D convolutions, Transformer-based methods with global attention mechanisms (Arnab et al., 2021; Zhang et al., 2021; Li et al., 2022) have achieved promising performance. These models can capture long-range dependencies across frames, enabling them to better understand the complex temporal relationships inherent in video data. While we aim to capture full spatial-temporal dynamics using Mamba architecture for more efficient video understanding.

**Image Foundation Models Adaptation.** Recently, image foundation models (IFMs) has made remarkable progress. The introduction of self-supervised learning (Oquab et al., 2023; Chen et al., 2020) and multi-modal contrastive learning (Jia et al., 2021; Radford et al., 2021; Zhai et al., 2023) techniques enable models to learn effective representations from unlabeled images. And, the come up of large-scale datasets such as LAION-5B (Schuhmann et al., 2022) and COYO-700M (Byeon et al., 2022) has led to even more powerful visual representations. IFMs trained on such web-scale datasets have been shown to be well generalized and effective in a wide range of computer vision tasks. Adapting these IFMs to these tasks has earned significant attention. Several approaches have been proposed to adapt IFMs to downstream tasks like detection (Kuo et al., 2022; Gu et al., 2021b; Minderer et al., 2023; Zhong et al., 2022), segmentation (Ma et al., 2022; Yang et al., 2024b; Ma et al., 2023b; Liu et al., 2023; Ma et al., 2025), image captioning (Hessel et al., 2021; Nukrai et al., 2022; Mokady et al., 2021), 2D/3D generation and reconstruction (Shi et al., 2025; Li et al., 2024c; Xiong et al., 2025; Li et al., 2024a; Ma et al., 2023a), and some trustworthy scenarios (Han et al., 2025; Yang et al., 2023b; 2021). In this work, we follow the research direction of EVL (Lin et al., 2022b), ST-Adapter (Pan et al., 2022), AIM (Yang et al., 2023a) and DiST (Qing et al., 2023), and aim to adapt IFMs to video understanding tasks.

**State Space Model.** Compared to Transformers that based on quadratic-complexity attention, State Space Models (SSMs) (Gu et al., 2021a) excels at processing long sequences with linear complexity. And recently, Mamba (Gu & Dao, 2023) distinguishes itself by incorporating a data-dependent selection mechanism and hardware-efficient algorithms, further improve its efficiency for sequential model-

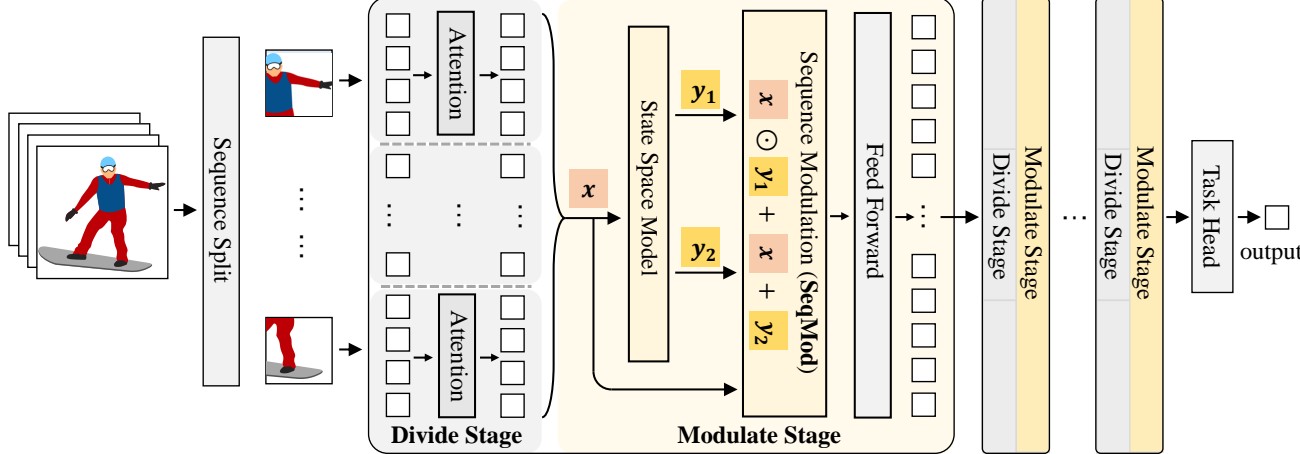

*Figure 1.* Overview of our proposed **MoMa**. The **Divide** stage aim to cut down the computational cost by narrowing the attention range. We utilize the original CLIP attention layers, but splits the input video sequence into smaller windows and processes attention independently for each window. In **Modulate** stage, we aim to capture full spatial-temporal dynamics using lightweight Mamba. We first forward the sequence through an SSM layer to obtain sequences of scale and bias, then use our proposed sequence modulation operation (**SeqMod**) to inject spatial-temporal information into IFMs without interfering with their pre-trained parameters. Finally, the output is fed into CLIP's feed-forward layer. Only SSM layers are trainable through the whole architecture.

ing tasks. In vision domain, backbones built on Mamba (Liu et al., 2024; Zhu et al., 2024; Hatamizadeh & Kautz, 2024) shows great potential, and introduces new solution in processing visual data at large scale and high dimension such as video (Li et al., 2024b; Chen et al., 2024) and 3D point cloud (Liang et al., 2024; Zhang et al., 2024). The great potential of Mamba motivates a series of works (Teng et al., 2024; Wan et al., 2024; Yang et al., 2024c;a; Pan et al., 2024) on downstream tasks, further demonstrates Mamba's better performances and higher GPU efficiency.

Unlike previous works which train Mamba-related architectures from scratch, we aim to build a hybrid architecture on top of pre-trained IFMs. This approach allows us to inherit the linear complexity advantages of Mamba while fully leveraging the powerful representations learned by IFMs.

**Mamba and Attention Hybrid Architectures.** Considering the efficiency of the Mamba structure, many methods have tried to combine Mamba with other structures. For example, Jamba (Lieber et al., 2024) and Mamba-2-Hybrid (Waleffe et al., 2024) attempt to construct hybrid language models by interleaving attention, MLP and Mamba layers. For vision tasks, MambaVision (Hatamizadeh & Kautz, 2024) combines Mamba, attention and convolution layers together. PoinTramba (Wang et al., 2024) utilizes both Transformer and Mamba encoders. However, these methods aim to find a scratch training model, and thus focus more on architecture design such as the optimal attention-Mamba ratio. In contrast, adapting Mamba to a pre-trained model has many constraints. Our architecture builds upon

CLIP and cannot change drastically. Thus, our focus is on how to maximize the advantages of Mamba without disrupting the original pre-training weight.

## 3. Method

### 3.1. Preliminary and Overview

This work adopts image foundation models (IFMs) for video understanding tasks. We leverage Mamba as an efficient adapter, due to its linear complexity in long-term spatial-temporal modeling. For IFMs, following previous works (Pan et al., 2022; Lin et al., 2022b), we use CLIP as a representative because of its rich semantic information, strong generalization, and multi-modal learning capabilities. Below, we first provide some preliminaries, and then present our architecture overview based on CLIP.

**CLIP.** Given a image $\mathbf{I} \in \mathbb{R}^{H \times W \times 3}$, CLIP first employs a patch embedding layer to encode the image into an initial image embedding, denoted as $\mathbf{I}^0$. CLIP then consists of a series of Transformer layers, where each layer performs two primary operations: Attention and FFN. Specifically, for the $i$-th layer with input feature $\mathbf{I}^i$, the forward process can be written as:

$$\mathbf{I}^{i+1} = \text{FFN}(\text{Attention}(\mathbf{I}^i)). \qquad (1)$$

Since CLIP is primarily an image model, it lacks inherent modeling of temporal information. Common approaches to adapting CLIP for video tasks typically involve adding additional temporal modules to capture the sequential dynamics.

**Architecture Overview.** Our method achieves parameter-efficient fine-tuning of CLIP by inserting a small number of learnable parameters. Specifically, for each Transformer layer, we introduce two important stage: **Divide** and **Modulate**, as shown in Figure 1. **(1)** In **Divide** stage, we aim to cut down the computational cost by narrowing the attention range in pre-trained CLIP. We achieve this by replacing the original image-level attention into a non-overlapping 2D window-level attention on video sequence. Using this method, we can reduce the computational cost while still able to capture short-term dependencies. **(2)** In **Modulate** stage, we aim to introduce full spatial-temporal interaction and capture long-range dependency. We utilize the linear-time complexity State Space Model (SSM) for efficient long sequence processing. Then, to integrate SSM's output with the original CLIP output, we propose a sophisticated sequence modulation operation to combine sequences from different sources effectively. Formally, the forward process in Equation (1) is modified as:

$$\mathbf{V}^{i+1} = \texttt{FFN}(\textbf{Modulate}(\textbf{Divide}(\mathbf{V}^i))), \quad (2)$$

where $\mathbf{V}^i \in \mathbb{R}^{HWT \times C}$ denotes the input video feature in the $i$-th layer. In the following sections, we will first detail the **Divide** stage (Section 3.2) and the **Modulate** stage (Section 3.3). Then, Section 3.4 describes the training process. And we provide an in-depth discussion in Section 3.5.

### 3.2. Divide Stage

Similar to self-attention, our **Divide** stage transforms the input video sequence without altering its shape. Since CLIP model is initially trained for image level understanding, most previous adapter based methods tend to follow this pattern and utilize CLIP attention to only process videos frame-by-frame (Pan et al., 2022; Yang et al., 2023a; Qing et al., 2023). While we aim to have a more precise control over the reception field of attention mechanism, and reduce the computational overhead accordingly. Thus, we propose to further divide each frame into multiple windows and apply local attention within each window. For input video feature $\mathbf{V}^i \in \mathbb{R}^{HWT \times C}$ at the $i$-th layer, we cut it with a fixed 2D window size $w \times w$. Let the total number of window across the video be denoted as $N$, so that:

$$\mathbf{V}^i \rightarrow [\mathbf{s}_1^i, \mathbf{s}_2^i, \ldots, \mathbf{s}_N^i],$$
$$\mathbf{s}_n^i \in \mathbb{R}^{w^2 \times C}, \quad N = HWT/w^2. \quad (3)$$

Here $\mathbf{s}_1, \mathbf{s}_2, \ldots, \mathbf{s}_N$ are non-overlapping flattened windows in frames of the video. For each window $\mathbf{s}_n^i$, we apply self-attention independently:

$$\mathbf{s}_n^i{}' = \texttt{Attention}(\mathbf{s}_n^i). \quad (4)$$

$\texttt{Attention}$ is the pre-trained CLIP attention layer. This attention mechanism allows the video sequence to exchange information locally inside each window. After attention operation, we concatenate the output of each window to obtain a new sequence for next stage process. The complete **Divide** stage can be written as:

$$\begin{aligned}
\mathbf{x}^i &= \texttt{Divide}(\mathbf{V}^i) \\
&= \texttt{Divide}([\mathbf{s}_1^i, \ldots, \mathbf{s}_N^i]) \\
&= [\texttt{Attention}(\mathbf{s}_1^i), \ldots, \texttt{Attention}(\mathbf{s}_N^i)] \\
&= [\mathbf{s}_1^i{}', \ldots, \mathbf{s}_N^i{}'].
\end{aligned} \quad (5)$$

**Complexity Analysis.** For a video sequence with spatial and temporal $H \times W \times T$, the computational complexity for full spatial-temporal attention is $O\left((HWT)^2\right)$. And the complexity for frame-by-frame spatial attention is $O\left((HW)^2T\right)$. For our **Divide** stage with window size $w \times w$, the complexity can be written as:

$$O\left(\frac{HWT}{w^2} \cdot (w^2)^2\right) = O\left(w^2 \cdot HWT\right). \quad (6)$$

By restricting the attention range within each window, we reduce the computational burden by a large margin and achieve linear time complexity.

### 3.3. Modulate Stage

In this stage, we aim to capture full spatial-temporal dynamics using Mamba, thus achieving linear-time complexity.

**Overview.** The main part of this stage is `SeqMod`, a novel sequence modulation operation to inject lightweight Mamba features into pre-trained Transformer-based IFMs. Rather than directly modifying the features of IFMs, `SeqMod` can adjust features by learnable scale and bias sequences.

In this stage, for any input sequence $\mathbf{x}^i$, we first use one SSM layer introduced in Mamba (Gu & Dao, 2023) to learn two sequences representing the scale and bias for modulation parameter. Then we conduct `SeqMod` operation and inject full spatial-temporal information during the process. We will explain in detail below.

**SSM Forwarding Layer.** We use an SSM layer to introduce full spatial-temporal interaction and capture long-range dependency. Figure 2 shows the detailed structure of our SSM forwarding layer. We use the similar architecture as proposed in Mamba (Gu & Dao, 2023). To extend the module's ability to output two sequences, for Divide stage output $\mathbf{x}^i$, we slightly modify the original SSM layer to double the channel number of the output projection layer. After that, we split the output by channel to obtain two output sequences $\mathbf{y}_1^i$ and $\mathbf{y}_2^i$. This can be formulated as:

$$\mathbf{y}_1^i, \mathbf{y}_2^i = \text{SSM}(\mathbf{x}^i). \quad (7)$$

Besides, following VideoMamba (Li et al., 2024b), we also conduct multiple times of bidirectional scanning operations through both spatial and temporal dimensions.

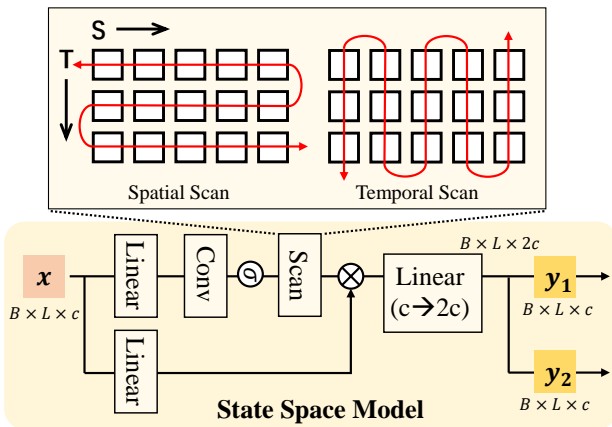

*Figure 2.* Detailed architecture of State Space Model (SSM) forwarding layer. The SSM module is designed to learn two sequences $\mathbf{y_1}$ and $\mathbf{y_2}$ that are used in further modulation operation. We conduct multiple times of bidirectional scanning through both spatial and temporal dimensions. The final projection channel is then doubled, and we split the output into two sequences.

**Sequence Modulation Operation (`SeqMod`).** For attention output $\mathbf{x}^i$ and SSM output $\mathbf{y}_1^i, \mathbf{y}_2^i$, we aim to effectively integrate these sequences together. We draw inspiration from the adaptive normalization (AdaN), and design a new sequence modulation operation, `SeqMod`. In the following, we will detail the design of our sequence modulation operation.

The concept of adaptive normalization (AdaN) refers to a series of methods that adjust the input's mean and variance *globally* with an outer condition. This concept was first introduced in FiLM (Feature-wise Layer Modulation) (Perez et al., 2017). At the same time, Huang & Belongie (2017) implemented this adaptation before the network normalization layer, which led to the term "Adaptive Instance Normalization" (AdaIN). Dumoulin et al. (2018) summarized these concepts as "feature-wise transformations", which have since been widely applied across a variety of tasks such as image recognition and generative modeling. For example, StyleGAN (Karras et al., 2018) uses it to inject style information into images, while DiT (Peebles & Xie, 2022) applies it to achieve text-to-image diffusion task. The general AdaN formula can be written as follows:

$$\text{AdaN}(\mathbf{x}) = \alpha_c \cdot \Phi(\gamma_c \mathbf{x} + \beta_c) + \mathbf{x}, \qquad (8)$$

Here, $\mathbf{x}$ is the input sequence. $\Phi(\cdot)$ is an arbitrary module, and $\alpha_c, \beta_c, \gamma_c$ are learnable *scaler* parameters derived from the given condition. Without loss of generality, here we choose $\Phi(\mathbf{x}) = \mathbf{x}$ for simplicity. Then the derived form is:

$$\text{AdaN}(\mathbf{x}) = \underbrace{\alpha_c \gamma_c}_{\text{scale}} \cdot \mathbf{x} + \underbrace{\alpha_c \beta_c}_{\text{bias}} + \underbrace{\mathbf{x}}_{\text{skip connection}}. \qquad (9)$$

We can observe from Equation (9) that the core of AdaN consists of three components: scale and bias, and skip connection. *Note that*, both the scale and bias are *scalar*, and the sequence is modulated *globally*. This global modulation best suits in tasks like style transfer, where image feature should be considered as a whole. However, in our scenario, the spatial-temporal information is fine-grained. Condensing the whole sequence output from SSM into a single scalar will obviously lose information. Therefore, we introduce a **seq**uence **mod**ulation operation: `SeqMod`. It extends the scalar scale and bias item into tensors with the same shape as the input sequence, and thus provide a fine-grained sequence-to-sequence modulation. Formally, this operation takes a similar form to that in Equation (9), and can be expressed as:

$$\texttt{SeqMod}(\mathbf{x}, \mathbf{y_1}, \mathbf{y_2}) = \underbrace{\mathbf{y_1}}_{\text{scale}} \odot \mathbf{x} + \underbrace{\mathbf{y_2}}_{\text{bias}} + \underbrace{\mathbf{x}}_{\text{skip connection}},$$
$$(10)$$

where $\mathbf{y_1}$ and $\mathbf{y_2}$ are the output of SSM, serving as sequential scale and bias; $\odot$ is element-wise multiplication. The modulated sequence is then fed into the original CLIP feed-forward layer for further processing:

$$\mathbf{V}^{i+1} = \texttt{FFN}(\texttt{SeqMod}(\mathbf{x}^i, \mathbf{y}_1^i, \mathbf{y}_2^i)). \qquad (11)$$

And the output $\mathbf{V}^{i+1}$ is then input for the $(i+1)$-th layer of **Divide** stage.

### 3.4. Training Process

We average the output feature from final layer of CLIP model to obtain video representation $\hat{\mathbf{y}}_o$.

$$\hat{\mathbf{y}}_o = \text{Average}(\mathbf{V}^L). \qquad (12)$$

Here $L$ is the total number of CLIP layers.

All CLIP's parameters are kept frozen during training, and only newly introduced SSM layers are trainable. Besides the classification loss, we also add a CLIP distillation loss to keep CLIP's ability on zero-shot understanding and ensure the output feature space is not altered too much. The whole architecture is trained end-to-end.

### 3.5. Other Fusion Designs

In Figure 3, we show some other designs to integrate two sequences. (a) Weighted average. (b) Element-wise max pooling. (c) Concatenation in channel dimension. Unfortunately, the performance of these methods are suboptimal, evidenced in Table 5. Similarly, as evidenced in Yang et al. (2024c), Mamba-based module does not inherently fit Transformer well. It is likely that directly inserting a completely new Mamba framework into an already trained Transformer may confuse the model, especially when the Transformer

*Table 1.* Comparison on Kinetics-400 (Kay et al., 2017) dataset. Views are in the format of #frames×#temporal×#spatial. We compare with previous PEFT methods as well as full-parameter-fine-tuning methods. Best results are in bold.

| Methods | PEFT | Extra Data | GFLOPs | Param (M) | Tunable Param(M) | Top-1 | Top-5 | Views |
|---|---|---|---|---|---|---|---|---|
| MViT-B (Fan et al., 2021) | ✗ | | 4095 | 37 | 37 | 81.2 | 95.1 | 64×3×3 |
| TokenLearner-L/10 (Ryoo et al., 2021) | ✗ | | 48912 | 450 | 450 | 85.4 | 96.3 | 64×4×3 |
| MViTv2-L (312 ↑) (Li et al., 2022) | ✗ | | 42420 | 218 | 218 | 86.1 | 97.0 | 32×3×5 |
| UniFormer-B (Li et al., 2021) | ✗ | IN-1k | 3108 | 50 | 50 | 83.0 | 95.4 | 32×4×3 |
| ViViT-L/16×2 FE (Arnab et al., 2021) | ✗ | IN-21k | 3980 | 311 | 311 | 80.6 | 92.7 | 32×1×1 |
| TimeSformer-L (Bertasius et al., 2021) | ✗ | IN-21k | 7140 | 121 | 121 | 80.7 | 94.7 | 64×1×3 |
| VideoSwin-L (Liu et al., 2022) | ✗ | IN-21k | 7248 | 197 | 197 | 83.1 | 95.9 | 32×4×3 |
| MTV-L (Yan et al., 2022) | ✗ | IN-21k | 18050 | 876 | 876 | 84.3 | 96.3 | 32×4×3 |
| VideoMamba-M (Li et al., 2024b) | ✗ | CLIP-400M | 2424 | 74 | 74 | 83.4 | 95.9 | 16×4×3 |
| ActionCLIP (Wang et al., 2021) | ✗ | CLIP-400M | 16890 | 142 | 142 | 83.8 | 97.1 | 32×10×3 |
| X-CLIP-L/14 (Ni et al., 2022) | ✗ | CLIP-400M | 7890 | 420 | 420 | 87.1 | 97.6 | 8×4×3 |
| AIM ViT-B/16 (Yang et al., 2023a) | ✓ | CLIP-400M | 1214 | 97 | 11 | 84.5 | 96.6 | 16×3×1 |
| DiST ViT-B/16 (Qing et al., 2023) | ✓ | CLIP-400M | 986 | 112 | 26 | 84.4 | 96.7 | 16×3×1 |
| **MoMa** (Ours) ViT-B/16 | ✓ | CLIP-400M | 451 | 97 | 11 | 83.7 | 96.5 | 8×3×1 |
| **MoMa** (Ours) ViT-B/16 | ✓ | CLIP-400M | 902 | 97 | 11 | **84.8** | **96.9** | 16×3×1 |
| EVL ViT-L/14 (Lin et al., 2022b) | ✓ | CLIP-400M | 8088 | 368 | 59 | 87.3 | - | 32×3×1 |
| AIM ViT-L/14 (Yang et al., 2023a) | ✓ | CLIP-400M | 5604 | 341 | 38 | 87.3 | 97.6 | 16×3×1 |
| DiST ViT-L/14 (Qing et al., 2023) | ✓ | CLIP-400M | 4534 | 343 | 40 | 87.6 | 97.8 | 16×3×1 |
| **MoMa** (Ours) ViT-L/14 | ✓ | CLIP-400M | 2076 | 342 | 39 | 86.7 | 96.7 | 8×3×1 |
| **MoMa** (Ours) ViT-L/14 | ✓ | CLIP-400M | 4152 | 342 | 39 | **87.8** | **98.0** | 16×3×1 |

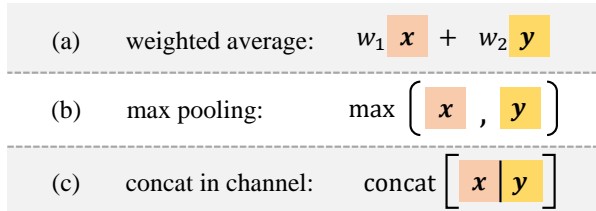

*Figure 3.* Some other designs of sequence fusion operations to integrate two sequences **x** and **y**. (a) Weighted average operation with hyperparameter $w_1$ and $w_2$; (b) Element-wise max pooling operation; (c) Concatenation in channel dimension.

parameters are frozen and unable to adapt to this new structure.

By drawing inspiration from multi-modal fusion techniques, specifically from AdaN, our **MoMa** introduce Mamba's information in a manner that minimally disrupts the original IFM's forwarding process while still effectively integrating Mamba's processing results. In essence, this method strikes a balance between the power of IFMs and the efficiency of Mamba's linear complexity, enabling effective video sequence understanding without the need to discard the benefits of prior pre-training weights.

## 4. Experiments

We performed a thorough evaluation of our model across several benchmarks: Section 4.1 for standard CLIP-adapter

baselines, Section 4.2 for long video baselines, and Section 4.3 for zero-shot transfer. Section 4.4 provides ablation studies to analyze our model from multiple perspectives.

**Implementation Details.** We use the pre-trained CLIP as our base model. We set the split window size $w = 8$. For SSM hyper-parameters, we set its hidden state 16, hidden dimension 384 and use gelu activation layer similar with CLIP. We adopt the same prompt in ActionCLIP (Wang et al., 2021). We use 8 Tesla V100 GPUs and fp16 precision for training. We use AdamW optimizer with learning rage 3e-4 and weight decay 0.05. Training a model on K400 dataset with 30 epochs takes about 12 hours to converge.

### 4.1. Standard Video Recognition Benchmarks

**Datasets.** We first evaluate our method on Kinetics-400 (K400) (Kay et al., 2017) and Something-Something V2 (SSv2) (Goyal et al., 2017). K400 is an action recognition dataset containing 240K training videos and 20K validation videos for 400 human action categories. Each video is trimmed to have a length of 10 seconds. SSv2 is a more challenging dataset requiring stronger temporal modeling (Sevilla-Lara et al., 2021). It contains about 168.9K training videos and 24.7K validation videos for 174 classes.

Table 1 and Table 2 show the comparison of our method with the state-of-the-art methods on K400 and SSv2. Notably, we achieve the best performance while requiring substantially fewer FLOPs (**25.6%** off than AIM and **8.5%** off than DiST) and trainable parameters than most other approaches.

*Table 2.* Comparison on Something-Something-v2 (Goyal et al., 2017) dataset. Views are in the format of #frames×#temporal×#spatial. We compare with previous PEFT methods as well as full-parameter-fine-tuning methods. Best results are in bold.

| Methods | PEFT | Extra Data | GFLOPs | Param (M) | Tunable Param(M) | Top-1 | Top-5 | Views |
|---|---|---|---|---|---|---|---|---|
| MViT-B (Fan et al., 2021) | ✗ | | 510 | 37 | 37 | 67.1 | 90.8 | 32×1×3 |
| MViTv2-B (Li et al., 2022) | ✗ | | 675 | 51 | 51 | 70.5 | 92.7 | 40×1×3 |
| MViTv2-L (312 ↑) (Li et al., 2022) | ✗ | | 8484 | 213 | 213 | 73.3 | 94.1 | 32×1×3 |
| UniFomer-B (Li et al., 2021) | ✗ | IN-1k | 777 | 50 | 50 | 71.2 | 92.8 | 32×1×3 |
| ViViT-L/16×2 (Arnab et al., 2021) | ✗ | IN-21k | 11892 | 311 | 311 | 65.4 | 89.8 | 16×4×3 |
| TimeSformer-L (Bertasius et al., 2021) | ✗ | IN-21k | 7140 | 121 | 121 | 62.4 | - | 64×1×3 |
| VideoSwin-B (Liu et al., 2022) | ✗ | IN-21k | 963 | 89 | 89 | 69.6 | 92.7 | 32×1×1 |
| MTV-B (Yan et al., 2022) | ✗ | IN-21k | 4790 | 310 | 310 | 67.6 | 90.4 | 32×4×3 |
| VideoMamba-M (Li et al., 2024b) | ✗ | CLIP-400M | 1212 | 74 | 74 | 71.0 | 92.7 | 16×2×3 |
| EVL ViT-B/16(Lin et al., 2022b) | ✓ | CLIP-400M | 2047 | 182 | 86 | 62.4 | - | 32×1×3 |
| AIM ViT-B/16 (Yang et al., 2023a) | ✓ | CLIP-400M | 2496 | 100 | 14 | 69.1 | 92.2 | 32×1×3 |
| DiST ViT-B/16 (Qing et al., 2023) | ✓ | CLIP-400M | 1972 | 112 | 26 | 70.9 | 92.1 | 32×1×3 |
| **MoMa** (Ours) ViT-B/16 | ✓ | CLIP-400M | 902 | 97 | 11 | 69.3 | 91.8 | 16×1×3 |
| **MoMa** (Ours) ViT-B/16 | ✓ | CLIP-400M | 1804 | 97 | 11 | **71.5** | **92.9** | 32×1×3 |
| EVL ViT-L/14(Lin et al., 2022b) | ✓ | CLIP-400M | 9641 | 484 | 175 | 66.7 | - | 32×1×3 |
| AIM ViT-L/14 (Yang et al., 2023a) | ✓ | CLIP-400M | 11508 | 354 | 50 | 70.6 | 92.7 | 32×1×3 |
| DiST ViT-L/14 (Qing et al., 2023) | ✓ | CLIP-400M | 9068 | 343 | 40 | 73.1 | 93.2 | 32×3×1 |
| **MoMa** (Ours) ViT-L/14 | ✓ | CLIP-400M | 4152 | 342 | 39 | 72.2 | 92.8 | 16×1×3 |
| **MoMa** (Ours) ViT-L/14 | ✓ | CLIP-400M | 8304 | 342 | 39 | **73.8** | **93.6** | 32×1×3 |

Specifically, on SSv2 dataset where the temporal modeling is more critical, our method outperforms the previous adapter methods like AIM and EVL by a large margin, and is also better than DiST. But DiST introduces a totally new encoder to capture the temporal information, while our method only needs a few adaptation layers inside the Transformer.

### 4.2. Long-term Video Recognition Benchmark

**Datasets.** To further demonstrate the effectiveness of our method in capturing long video sequence, we evaluate our method on Breakfast (Kuehne et al., 2014) and COIN (Tang et al., 2019). Breakfast comprises 1,712 videos, encapsulating 10 intricate cooking activities over 77 hours. COIN features 11,827 videos across 180 unique procedural tasks, with an average duration of 2.36 minutes. Following Video-Mamba's (Li et al., 2024b) setting, we further PEFT our models trained on K400 from Table 1.

Table 3 shows the comparison result. Our method surpasses those non end-to-end methods by a large margin, demonstrating the effectiveness of our method in capturing long-term video understanding. Meanwhile, by conducting PEFT on top of CLIP model, we also outperforms VideoMamba.

### 4.3. Zero-shot Ability

Similar to DiST (Qing et al., 2023), our method also has the ability to perform zero-shot tasks. Following DiST's setting, we use the model trained on K400, and evaluate it on two relatively small video recognition datasets: HMDB51 (Kuehne et al., 2011) and UCF101 (Soomro et al., 2012). Table 4

shows the comparison result. Our method surpasses DiST on both datasets, demonstrating the effectiveness of our method in zero-shot transfer learning.

### 4.4. Ablation Studies

In this section, we introduce some baseline implementations under similar Mamba architecture. We also ablated each crucial part of our architecture. All ablations are conducted on the Kinetics-400 dataset and CLIP ViT-B/16 backbone with 16 frames as input, unless otherwise specified.

**Baselines for Sequence Modulation Operation.** As discussed in Section 3.5, inserting Mamba module directly into well pre-trained Transformer architecture is sub-optimal. Here we compare our method with other implementations to integrate the two sequences. (1) **Skip**: We skip the whole SSM forward module and use sequence from `Divide` operation directly, which can be seen as a lower bound. (2) **Add**: We add the two sequences in an 1:1 ratio; (3) **Max**: Replacing addition with element-wise max introduces non-linearity, and this may enhance the model capability; (4) **Concat**: We concatenate the two sequences in channel and then use a linear layer to reduce the dimension; (5) **Raw-AdaN**: We use the vanilla AdaN formula which learns *scaler* scale and bias.

Table 5 shows the comparison result. We find that simple fusion methods such as **Add** and **Max** performs even worse than purely skip the whole SSM block. This indicates fusing different output sequences directly may lead to information "confusion", especially when the two features

*Table 3.* Comparison on long video datasets Breakfast (Kuehne et al., 2014) and COIN (Tang et al., 2019). $f_{32}$ and $f_{64}$ denote the number of frames sampled during training and testing. Best results are in bold.

| Model | end-to-end | Archi | Extra Data | Breakfast | COIN |
|---|---|---|---|---|---|
| Timeception (Hussein et al., 2019a) | ✗ | 3D-ResNet Conv. | IN-1K+K400 | 71.3 | - |
| VideoGraph (Hussein et al., 2019b) | ✗ | I3D Conv.+Atten. | IN-1K+K400 | 69.5 | - |
| GHRM (Zhou et al., 2021) | ✗ | I3D Graph Conv. | IN-1K+K400 | 75.5 | - |
| Distant Supervision (Lin et al., 2022a) | ✗ | TimeSformer Atten. w/ KB | IN-21K+HTM | 89.9 | 90.0 |
| ViS4mer (Islam & Bertasius, 2022) | ✗ | Swin-B SSM | IN-21K+K600 | 88.2 | 88.4 |
| Turbo$_{f32}$ (Han et al., 2022) | ✓ | VideoMAE-B | K400 | 86.8 | 82.3 |
| Turbo$_{f32}$ (Han et al., 2022) | ✓ | VideoMAE-B | K400+HTM-AA | 91.3 | 87.5 |
| VideoMamba$_{f32}$ (Li et al., 2024b) | ✓ | VideoMamba-M | K400 | 94.8 | 88.3 |
| VideoMamba$_{f64}$ (Li et al., 2024b) | ✓ | VideoMamba-M | K400 | 95.8 | 89.5 |
| **MoMa** (Ours)$_{f32}$ | ✓ | CLIP ViT-L/14 | K400+CLIP-400M | 95.1 | 89.0 |
| **MoMa** (Ours)$_{f64}$ | ✓ | CLIP ViT-L/14 | K400+CLIP-400M | **96.9** | **90.0** |

*Table 4.* Comparison of **zero-shot** action recognition performance on HMDB51 (Kuehne et al., 2011) and UCF101 (Soomro et al., 2012) datasets. All models are based on CLIP ViT pre-training.

| Method | Model | HMDB51 | UCF101 |
|---|---|---|---|
| ActionCLIP(Wang et al., 2021) | B/16 | 40.8 | 58.3 |
| X-CLIP (Ni et al., 2022) | B/16 | 44.6 | 72.0 |
| DiST (Qing et al., 2023) | B/16 | 55.4 | 72.3 |
| DiST (Qing et al., 2023) | L/14 | 57.5 | 74.9 |
| **MoMa** (Ours) | B/16 | 56.2 | 74.0 |
| **MoMa** (Ours) | L/14 | **59.1** | **76.2** |

*Table 6.* **Evaluation of various window sizes during the Divide stage.** In addition to the 2D window, we also evaluate a 3D window with dimensions $4 \times 4 \times 4$. Increasing the window size does not always bring positive gain.

| Window Size | Frame/Sec | Attn(%) | FFN(%) | Top-1 |
|---|---|---|---|---|
| Full ($40 \times 30$) | 8.2 | 39.7 | 41 | 69.2 |
| $16 \times 16$ | 10.4 | 31.7 | 52 | 70.3 |
| $8 \times 8$ (Ours) | 14.0 | 23.9 | 70 | 70.1 |
| $4 \times 4 \times 4$ (3D) | 14.0 | 23.9 | 70 | 68.2 |
| $4 \times 4$ | 15.1 | 18.8 | 75 | 67.5 |

*Table 5.* **Comparison of different sequence modulation operations.** We implement three baseline methods: "Add", "Max" and "Concat" as mentioned in Figure 3. "Skip" is also provided as a lower bound. Our operation "`SeqMod`" outperforms all baselines.

| | Methods | Operation | Top-1 | Top-5 |
|---|---|---|---|---|
| (1) | Skip | $x$ | 72.4 | 90.8 |
| (2) | Add | $x + y$ | 69.3 | 88.7 |
| (3) | Max | $\max(x, y)$ | 70.2 | 89.6 |
| (4) | Concat | $\text{Linear}([x, y])$ | 75.5 | 92.5 |
| (5) | Raw-AdaN | $\alpha_y \cdot x + x + \beta_y$ | 78.9 | 94.1 |
| (6) | **`SeqMod`** | $y_1 \odot x + x + y_2$ | **84.8** | **96.5** |

are misaligned or have inconsistent distributions. While "Concat" preserves more information by adding learnable linear layer, it does not guarantee that the modification of the sequence is orthogonal. **Raw-AdaN**, on the other hand, uses a "soft gating" mechanism, and changes to the sequence's features are orthogonal to the original CLIP feature outputs. This helps avoid altering the feature distribution of the original CLIP too much, reducing interference with pre-trained knowledge. Built on top of this mechanism, our "`SeqMod`" operation shows powerful sequence fusion capabilities, far surpassing all baselines.

**Window Size in Divide Operation.** We here explore the impact of window size in the **Divide** operation on SSv2 dataset. We upscale the video resolution to $640 \times 480 \times 16$.

Thus, the input feature scale is $40 \times 30 \times 16$ (patches). Beside 2D window, we also explore 3D window.

Table 6 shows the comparison result. The divide operation brings both **speed up** and **performance gain**. **Speed**: Divide the window from full to $8 \times 8$ brings significant increase in speed (175% fps) since attention operation dominates the computation. However, further dividing brings marginal improvement when FFN becomes the bottleneck. **Performance**: Conducting full attention on the whole image performs worse than window dividing. That's may because CLIP originally trained on attention sequence. Understanding long sequence is out of its training scope.

**Different Layer Design Pattern.** We offer several design of architectural patterns below: (1) [TM]12: Alternating sequence of Transformer and SSM layers, each repeated 12 times (our current design). (2) [T]12[M]12 : 12 consecutive Transformer layers followed by 12 SSM layers, with the SSMs functioning as the decoder rather than being inserted as an adapter in the middle of the encoder. (3) [T]6[TM]6 : No modulation for the first half of the Transformer layers. (4) [TTM]6 : One modulation layer is inserted between every two transformer layers

Table 7 shows the result. We find that: (1) Adapting only the latter half of the backbone is sub-optimal, Since learn to recognizing temporal information in the early stages of

*Table 7.* **Ablation study on various layer design patterns.** We found that the best performance is achieved by uniformly alternating between transformer and mamba layers.

|  | Pattern | Top-1 | Top-5 |
|---|---|---|---|
| (1) | [TM] 12 **(Ours)** | **83.7** | **96.5** |
| (2) | [T] 12 [M] 12 | 80.6 | 92.8 |
| (3) | [T] 6 [TMM] 6 | 81.5 | 94.0 |
| (4) | [TTMM] 6 | 82.8 | 95.5 |

video encoding is essential for effective processing. (2) The [TTMM]6 pattern performs worse than [TM]12 , suggesting that a more balanced integration of both components in the adapter structure is more beneficial.

### 4.5. Speed Comparison

We compare the speed performance of **MoMa** with two classic video sequence processing approaches: full-attention and spatial-temporal separation. Specifically, we select representative methods from each category. UMT (Li et al., 2023) uses full-attention, treating the entire video sequence as a single input, leading to quadratic complexity and high computational cost. AIM (Yang et al., 2023a), on the other hand, applies spatial and temporal attention separately, first at the spatial level and then at the temporal level.

As shown in Figure 4, the GPU memory usage of UMT increases rapidly with the number of input frames and eventually runs out of memory at 32 frames. Its inference FPS also drops significantly as the frame count rises. In contrast, both AIM and our method show stable memory growth and a gradual decrease in inference speed as the number of frames increases. However, thanks to our powerful **SeqMod** operation, our method achieves faster processing and experiences a more gradual increase in memory consumption. Additionally, the SSM module is more parameter-efficient than the attention module, further enhancing the overall efficiency of our approach.

### 5. Conclusion

In this work, we proposed a novel method to adapt IFMs for video understanding. We first introduces a modulation operation **SeqMod**. It injects spatial-temporal information into the features of IFMs without interfering with their pre-trained parameters, thereby maintaining computational efficiency. We further proposed our architecture **MoMa** with a "Divide-and-Modulate" strategy. It applies local attention within each video frame, followed by a modulation step to capture the full spatial-temporal dynamics. Through extensive experiments on multiple video understanding benchmarks, we demonstrate that our approach **MoMa** improves both performance and efficiency compared to existing methods, providing a balance between accuracy and scalability.

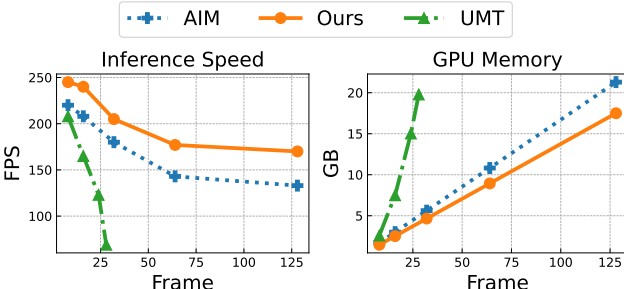

*Figure 4.* **Comparison of speed performance.** Compared with full attention method UMT and spatial-temporal separate method AIM, our method, powered by the **SeqMod** operation, achieves faster processing with a more gradual increase in memory usage.

### Impact Statement

This paper presents work aimed at advancing video understanding by improving the spatial-temporal modeling efficiency of pre-trained image foundation models. Our method has the potential to enhance various applications, such as video analytics in healthcare, security, and autonomous systems. While there are broad societal implications, including privacy concerns and ethical considerations in video-based AI, we do not believe any specific issues need to be highlighted beyond the general ethical discussions in the field of machine learning.

### Acknowledgement

This work is supported by the National Key R&D Program of China (No. 2022ZD0160703), National Natural Science Foundation of China (No. 62306178) and STCSM (No. 22DZ2229005), 111 plan (No. BP0719010).

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
