# OpenReview forum: "MoMa: Modulating Mamba for Adapting Image Foundation Models to Video Recognition"
_ICML.cc/2025/Conference — ICML 2025 poster_

### Official Review · Reviewer_17be · 2025-02-16

**Overall Recommendation:** 3

**Summary:**

The method in the paper attempts to adapt image foundation models for video understanding tasks. The authors introduce MoMa, an efficient adapter framework that integrates Mamba’s selective state space modeling into image foundation models. They propose a novel SeqMod operation designed to inject spatial-temporal information into pre-trained image foundation models without disrupting their original features. MoMa is incorporated into a Divide-and-Modulate architecture, enhancing video understanding while maintaining computational efficiency. Extensive experiments on multiple video benchmarks demonstrate that MoMa achieves superior performance compared to existing models, with reduced computational costs.

**Claims And Evidence:**

Yes, the claims made in the submission are supported by clear and convincing evidence from the experimental results. The authors demonstrate that the MoMa framework, which integrates Mamba’s selective state space modeling into image foundation models.

**Essential References Not Discussed:**

No, the paper makes a good review of the literature. It discusses relevant background and builds upon existing work in the domain of state space models (SSM) and their application to video understanding.

**Experimental Designs Or Analyses:**

Yes, the experimental settings are clear and well-explained. However, the benchmark might be somewhat outdated or overly saturated for the current video understanding research.

**Methods And Evaluation Criteria:**

Yes, the proposed efficient method has the impact on application side.

**Other Comments Or Suggestions:**

NA

**Other Strengths And Weaknesses:**

Strength
1. This paper effectively adapts an image model to handle video tasks, a crucial step for video understanding research, especially since image models are often trained on much larger datasets.
2. The use of Mamba (or other SSM models) is promising, as it has demonstrated strong performance across various NLP and computer vision tasks. It is exciting to see this approach applied in the video understanding domain as well.

Weaknesses: See Qusetion section

**Questions For Authors:**

1. The divide and modulate stages are actually the combination of window-attn and SSM hybird architecture.  Are there any ablation studies focusing on the layer design? (e.g., adding the SSM layer only in certain parts of the architecture like in Jamba)?
2. Video understanding as a field has evolved rapidly. The benchmarks used in the paper seem to be outdated, particularly with datasets like K400 and SSv2, which are now saturated (with performance rates over 95%). Would it be possible to include more recent datasets such as ActivityNet, or VideoMME for a more challenging comparison?
3. For a fair comparison, it would be helpful if the authors could also include the input image size used in Table 1 and Table 2 like the table in VideoMamba, as it may influence the performance result.

**Relation To Broader Scientific Literature:**

The paper is an extension from the mamba and other SSM model used in NLP tasks.


The paper builds upon existing work in the domain of selective state space modeling (SSM), particularly in the context of natural language processing (NLP) tasks. It extends the Mamba architecture and similar SSM models to address video understanding challenges. While the approach is an extension, it would be helpful for the authors to more explicitly position this work within the broader literature and discuss how it advances beyond prior methods, particularly in video processing and multimodal tasks.

**Theoretical Claims:**

No, there is quite a little theoretical part in this paper. While the paper introduces a novel method for improving video recognition, it lacks a thorough theoretical explanation of the underlying mechanisms and the potential limitations of the approach.

---

> ### Author Rebuttal · Authors · 2025-04-01
>
> >## 1. Ablation studies focusing on the layer design
>
> Thanks for your advice!
>
> As discussed in Section 3.5, unlike Jamba, which focuses on fine-grained architectural design, our architecture builds upon CLIP and cannot undergo drastic changes. Instead, we focus on how to maximize the advantages of Mamba without disrupting the original pre-trained weights.
>
> Below, we supplement an ablation on architectural patterns.
>
> - ``[TM]12``: Alternating sequence of Transformer and SSM layers, each repeated 12 times (our current design).
> - ``[T]12[M]12``: 12 consecutive Transformer layers followed by 12 SSM layers, with the SSMs functioning as the decoder rather than being inserted as an adapter in the middle of the encoder.
> - ``[T]6[TM]6``: No modulation for the first half of the Transformer layers.
> - ``[TTM]6``: One modulation layer is inserted between every two transformer layers.
>
> | Pattern           | Top-1 | Top-5 |
> | ----------------- | ----- | ----- |
> | ``[TM]12`` (Ours) | 83.7  | 96.5  |
> | ``[T]12[M]12``    | 80.6  | 92.8  |
> | ``[T]6[TMM]6``    | 81.5  | 94.0  |
> | ``[TTMM]6``       | 82.8  | 95.5  |
>
> From the table we find that:
>
> - Using MoMa directly as the decoder performs the worst. Under this pattern, the encoder is entirely non-trainable, preventing it from adapting to understand video inputs. As a result, all temporal information must be learned within the lightweight SSM layer, which makes it difficult to capture temporal dependencies effectively.
> - Adapting only the latter half of the backbone is sub-optimal, Since learn to recognizing temporal information in the early stages of video encoding is essential for effective processing.
> - The ``[TTMM]6`` pattern performs worse than ``[TM]12``, suggesting that a more balanced integration of both components in the adapter structure is more beneficial.
>
> We will add this table into our final version.
>
> >## 2. The benchmark might be outdated
>
> Here we supplement a new experiment by equipping MoMa with MLLMs and answer complicated video understanding tasks.
>
> ### 2.1 Equipping MoMa into MLLMs
>
> **Experiment setting**
>
> We first train MoMa with CLIP-L backbone on Kinetics-700 for video understanding. Then we apply a VideoLLaMA2-style VLM architecture and train the only projection part on video instruction tuning dataset. We use Qwen2.5-1.5B LLM decoder and ``LLaVA-Video-178k`` for training. Finally, we evaluate on a subset (``[action_antonym, action_localization,action_sequence]``) of MVBench.
>
> - `action_antonym`: distinguish the correct action from two inversely ordered actions.
> - `action_localization`: determine the time period when a certain action occurs.
> - `action_sequence`: retrieve the events occurring before or after a specific action.
>
>
> Notably, our training configuration involves significantly fewer resources compared to contemporary VLMs. We use only 0.5M video data for pre-training and 178k video instruction tuning data for fine-tuning, only **5%** of the 12M+ video-text data used in VideoLLaMA. This resource-constrained setup inevitably limits the model's generalizability across comprehensive video understanding tasks. Therefore, we choose some action-related subtasks from MVBench since these are most similar with our training resource.
>
> | Model         | LLM          | AA   | AL   | AS   |
> | ------------- | ------------ | ---- | ---- | ---- |
> | LLaMA-Adapter | LLaMA-7B     | 51.0 | 21.5 | 23.0 |
> | LLaVA         | Vicuna-7B    | 63.0 | 20.5 | 28.0 |
> | VideoLLaMA    | Vicuna-7B    | 51.0 | 22.5 | 27.5 |
> | Ours          | Qwen2.5-1.5B | 52.1 | 22.7 | 23.4 |
> | GPT-4V        | GPT-4        | 72.0 | 40.5 | 55.5 |
>
> We achieve comparable performance with LLaVA and Video LLaMA while operating with merely **5%** of their training data and **21%** of their parameter budget (1.5B vs. 7B parameters), demonstrating our efficiency while showing our scaling potential across a broader range of wider application.
>
>
> >## 3. Include image size in Table 1 & 2
>
> Thanks for the suggestion. For a fair comparison, all input resolution in our Table 1 and Table 2 are 224, unless otherwise specified. (For example, the MViTv2-L (312$\uparrow$) in the sixth row (L284) of Table 1 indicates an input size of 312, which is larger than the default size.) We will clarify this in the final version.

---

> > ### Comment · Reviewer_17be · 2025-04-02
> >
> > Thank you for the detailed responses. I appreciate that the authors have conducted additional experiments to address my concerns, including the ablation on the layer design and the use of more recent datasets.
> >
> > However, I find the results of equipping MoMa with MLLMs somewhat underwhelming. While the resource-constrained setting is understandable, the performance gains appear limited, and it remains unclear whether MoMa offers a substantial advantage in more generalizable or large-scale settings.
> >
> > Considering the improved clarity and the effort to address all points raised, I am increasing my score to a 3 (weak accept), though I remain somewhat reserved about the broader impact of the proposed method.

---

> > > ### Author Response · Authors · 2025-04-05
> > >
> > > We apologize for the delayed response, as we are working on the MLLM experiments to scale up training data.
> > >
> > > We appreciate your concern regarding the generalizability of our approach. We would like to clarify that MoMa is designed as a parameter-efficient fine-tuning (PEFT) method, pursuing fast adaptation and efficient inference.
> > > From this perspective, we believe that the performance of MoMa is already quite competitive, especially considering the limited training data and resources (**1.5B vs 7B**, **5\%** training data). Besides, it's able to reduce nearly **30%** fewer FLOPs compared to other PEFT methods.
> > >
> > > As reviewer requested, we further conducted an experiment to validate the performance of MoMa in a more generalizable setting by equipping it with MLLMs.
> > >
> > >
> > > | Model         | LLM          | AA   | AL       | AS   |
> > > | ------------- | ------------ | ---- | -------- | ---- |
> > > | LLaMA-Adapter | LLaMA-7B     | 51.0 | 21.5     | 23.0 |
> > > | LLaVA         | Vicuna-7B    | 63.0 | 20.5     | 28.0 |
> > > | VideoLLaMA    | Vicuna-7B    | 51.0 | 22.5     | 27.5 |
> > > | Ours          | Qwen2.5-1.5B | 52.1 | 22.7     | 23.4 |
> > > | Ours  (10x training data)     | Qwen2.5-1.5B | **63.9** | **27.0** | **32.2** |
> > >
> > >
> > > We managed to provide the MLLM experiments by increasing the training data to 5M (10x of the original training data). The results are encouraging, as incorporating additional data can enhance our performance. We achieve SOTA in all three benchmarks, particularly in action localization (AL) and action sequence (AS) tasks, where temporal information processing is critical.
> > >
> > >
> > > Lastly, we are grateful for your recognition of our efforts and your commitment to raising our score.

---

### Official Review · Reviewer_XKgu · 2025-03-12

**Overall Recommendation:** 4

**Summary:**

This paper proposed a framework "modulated Mamba" to adapt image foundation models for video understanding tasks by PEFT. There are two stages within the frame work. The first stage is "divide", which runs CLIP feature extraction on the pacthes of each frame. The second stage is "SeqMod" which draws intuiation from AdaN and merge the output from SSM layer to integrate both spatial and temporal informtion. The proposed method achieved state-of-the-art results on both short and long video understanding benchmarks.


## update after rebuttal
The authors addressed my concern on the ablation analysis and related works, I increased score to accept and suggest adding these analysis in the final version.

**Claims And Evidence:**

yes. the rationale is clear for the proposed method.

**Essential References Not Discussed:**

no

**Experimental Designs Or Analyses:**

The experimental design is comprehensive. Though I would like to see more analysis on the ablation studies. For example, why does different merging method matters so much in Tab. 5? "concat" and "SeqMod" have the same input without extra parameters but there are alost 10% difference. I also have a question below.

Does the method applies to differen video understanding tasks like temporal grounding? For long video understanding, does it applies to longer videos like LVBench?

**Methods And Evaluation Criteria:**

I'm not sure if CLIP is the best model to use for extract the patch features, becasue, as the author also mentioned in the paper, CLIP is trained for full images. Tab. 6 shows how you extact patch features matters a lot, the authors might try different ways for get patch wise features, e.g. the pretrained tokenizer in ViT or even traditional methods that focus more on local features.

The methods and eval metrics are valid.

**Other Comments Or Suggestions:**

no

**Other Strengths And Weaknesses:**

As I said in other sections, the method is novel and the results are good, but more analysis are needed.

**Questions For Authors:**

I don't thin adaptive normalization (AdaN)  is proposed in Perez et al., 2017 (L218), wrong citation or you refers to FiLM? I think you may be referring to a group of method following similar formula, the authors should make it clear in the text and add to related works, since it's the core idea behind he scene.

In Tab.5, it’s counterintuitive that "add" and "max" perform worse than "skip", since the trained SSM layer should learn to minimize y and achieve the same results with skip?

**Relation To Broader Scientific Literature:**

The paper dives deep into the topic of using PEFT to integrate temporal information into a pretrained image foundatio model, and cited related works related to video understanding, image foundation models, PEFT and SSM.

**Theoretical Claims:**

yes

---

> ### Author Rebuttal · Authors · 2025-04-01
>
> >## 1. Whether CLIP is the best model
>
> Note that our MoMa is an adapter method and is thus backbone-agnostic. We choose CLIP as our backbone following previous methods DiST and AIM for fair comparison.
>
> | Method    | Accuracy |
> | --------- | -------- |
> | MoMa-CLIP | 83.7     |
> | MoMa-MAE  | 81.2     |
>
> Here we replace the CLIP backbone with ViT-MAE from facebook, considering that MAE-style pre-training may be more focused on image local feature. The performance drops since the MAE is pre-trained on ImageNet-1K, which is not as large as CLIP.
> Besides, discussing which image pre-trained model better suits video understanding is out of the scope of this paper.
>
> >## 2. Why different merging method matters so much. Explain Tab. 5 more.
>
> We’ve provided a new ablation study on our SeqMod operation. For more details, please refer to our response to ``R1(5nFH) #2: Further analysis on SeqMod operation``. We hope this clarifies and provides a more comprehensive understanding of SeqMod.
>
> >## 3. Temporal grounding & LVBench
>
> Thanks for the suggestion!
>
> We've add a new experiment by equipping MoMa with MLLMs and answer three complicated video understanding tasks from MVBench:
> - `action_antonym`: distinguish the correct action from two inversely ordered actions.
> - `action_localization`: determine the time period when a certain action occurs.
> - `action_sequence`: retrieve the events occurring before or after a specific action.
>
> We achieve comparable performance on these tasks compared with LLaVA and Video LLaMA while operating with merely 5% of their training data and 21% of their parameter budget (1.5B vs. 7B parameters), demonstrating our efficiency while showing our scaling potential across a broader range of wider application. Please see our response to ``R4(17be) #2.1 Equipping MoMa into MLLMs`` for implementation detail.
>
>
> >## 4. Related works
>
> We here supplement a detailed literature review.
>
> The concept of learning scale and bias for feature adaptation was first introduced in FiLM [1] (**F**eature-w**i**se **L**ayer **M**odulation) in 2017. At the same time, [2] implemented this adaptation before the network normalization layer, which led to the term ‘adaptive instance normalization’ (AdaIN). A 2018 survey [3] summarized these concepts as ‘feature-wise transformations,’ which have since been widely applied across a variety of tasks such as image recognition and generative modeling. Thus, AdaN refers to a family of methods that share this underlying principle, including, but not limited to, FiLM, AdaIN, and DiT.
>
> We will clarify the definition of AdaN in the final version for better understanding. Thank you for the suggestion, and please let us know if there are any additional references or concepts we should include.
>
> [1] FiLM: Visual Reasoning with a General Conditioning Layer
>
> [2] Arbitrary Style Transfer in Real-time with Adaptive Instance Normalization
>
> [3] https://distill.pub/2018/feature-wise-transformations

---

> > ### Comment · Reviewer_XKgu · 2025-04-06
> >
> > I appreciate the authors for the thorough response. My concerns have been addressed. The authors should add related clarifications to the final version, in particular the analysis on different merging method in Tab. 5. I increased my score to accept.

---

### Official Review · Reviewer_jWpG · 2025-03-12

**Overall Recommendation:** 3

**Summary:**

The paper proposes using Mamba layers as adapters to apply CLIP pre-trained transformer-based image models for video tasks. For each transformer block, the proposed method first divides each frame into multiple windows, applies self-attention within each window, then the tokens of all windows of all frames are concatenated into a flat sequence, over which Mamba layer is applied in multiple scanning orders. The output of Mamba is linearly projected into a 2x dimensional space and split into two to produce a scale and bias terms that are used to modulate the self-attention (image) features, taking inspiration from Adaptive Normalization works. Experiments are carried out on standard video classification datasets (Kinetics400, SSv2) and long-term video recognition datasets (COIN, Breakfast), and zero-shot transfer to smaller scale datasets (HMDB51, UCF101). Ablation experiments are presented to justify architectural choices, mainly the way the modulation of the image features is performed and the size of the window in the divide stage. The proposed method outperforms existing baselines on all datasets, while having reduced latency and memory footprint compared to spatio-temporal attention architectures.

**Claims And Evidence:**

The use of the Divide part (self-attention applied on sub-windows of frames) is not justified through compelling experiments. When using window (local) attention, the performance decreases (Table 6) for a small increase in throughput. Using higher resolution inputs with window attention could have revealed some interesting interplay between mixing of information through self-attention (larger windows) vs mixing directly over space and time through Mamba (smaller windows). With the current experiments, Divide only seems to complicate the story without adding convincing benefits.

For the SeqMod part, an ablation of the Mamba state size would be needed to see if you can get better performance by increasing the capacity of Mamba together or instead of using the proposed modulation.

**Essential References Not Discussed:**

NA

**Experimental Designs Or Analyses:**

The experiments on long-term action recognition (COIN, Breakfast) are confusing. The authors perform end-to-end finetuning (Table 3), but line 368 mentions PEFT? There are no implementation details about the long-term fine-tuning experiments.
What batch size is used in experiments? 12h to converge on K400 seems quite long, given that only the Mamba parameters need to be learnt.

The comparisons in Tables 1,2 could be misleading as they don’t mention pre-training data. It is not fair to directly compare the proposed method which uses CLIP weights trained on a very large dataset to e.g. ViViT that uses only Imagenet pretrained weights. This is included in Table 3, but still a bit misleading: the CLIP pre-training dataset should be included under Dataset.

Since the authors include fine-tuning results for long-term action recognition, the fine-tuning results for k400 and ssv2 should also be included.

**Methods And Evaluation Criteria:**

The evaluation focuses on video classification alone; it should be improved by adding a VLM task, given that CLIP has strong semantic features.

**Other Comments Or Suggestions:**

typos:
L100: without interfering the pre-trained IMF → with the
L55: significantly improvements → significant improvements?
L70: “While we aim to capture …” sentence needs rephrasing
L84: “be well generalized” – needs rephrasing
L154: Given a image → an image
L173: “While we aim to have a more precise…” sentence needs rephrasing
L319: rage → rate
L325: we fine-tune … and employs → employ
L428: we first introduces → introduce

**Other Strengths And Weaknesses:**

The writing needs improvement.

There are multiple typos, some that hinder comprehension (included in the section below). The notations are not all well explained or used consistently. Image dimensions H,W seem to represent pixels at L154, but the model probably operates on embedded patches. When using local attention, 16x16 windows are expressed in terms of pixels or patches?
L379 “Key frame number 16” – how are key frames extracted? There is a typo in this sentence as in L428 “frame number 16” ?
In Fig3, the notations are not clear to me, how do y1 and y2 relate to x,y in the figure?

First part of 3.5 talks about Related work again, would be better placed in Section 2.

**Questions For Authors:**

1. What is the performance when fine-tuning on K400 and SSv2?
2. Did the authors try to ablate the SSM state size and hidden size?

**Relation To Broader Scientific Literature:**

The discussion about SSM for vision should be more general and include hybrid SSM works prior to or concurrent to Mamba. ViS4mer (using S4 SSM) is included in experiments comparison, but it’s not discussed in related work. TranS4mer should be mentioned as well. They both rely on standard image encoders. More recently, TRecViT used pre-trained weights with linear recurrent units, which are very similar to Mamba layers.

**Theoretical Claims:**

NA

---

> ### Author Rebuttal · Authors · 2025-04-01
>
> >## 1. Divide part is not justified, using higher resolution inputs.
>
> Thanks for advice! Our original $224 * 224$ comparison aimed to align with other baselines, but we’ve now conducted additional ablation studies at $640*480$ resolution (SD video standard) to address your concern.
>
> **Experiment details**
> - Input video resolution $640 * 480 * 16$
> - CLIP ViT-B/16 backbone
>   - After image embedding layer, the feature shapes $40 * 30 * 16$ (patches).
> - Window size is defined on patch level.
>
> |Window Size|Speed(frame/s)|Attention(%)|SSM(%)|FFN(%)|Top-1(Acc)|
> |-|-|-|-|-|-|
> |Full| 8.2 |39.7|1.7|41|69.2|
> |16| 10.4|31.7|1.6|52|70.3|
> |8(Ours)| 14.0|23.9|2.1|70|70.1|
> |4| 15.1|18.8|2.0|75|67.5|
>
> (* Experiment conducted on SSv2. Speed tested on A100.)
>
> From the table we observe that the divide operation brings both **speed up** and **performance gain**.
> - **Speed**: Divide the window from full to $8*8$ brings significant increase in speed (175% fps) since attention operation dominates the computation. However, further dividing brings marginal improvement when FFN becomes the bottleneck.
> - **Performance**: Conducting full attention on the whole image performs worse than window dividing. That's may because CLIP originally trained on $16*16$ attention sequence. Understanding long sequence is out of its training scope.
>
> >## 2. Ablation of the Mamba state size
>
> I'm not sure whether you're asking for Mamba's state size or the model’s hidden dimension.
>
> For state size, we’ve set to 16 as default. Increasing the state size does not enhance the model’s capacity. Even large-scale models like falcon-mamba-7b ([1]) and Jamba (52B MoE) ([2]) use a state size of 16.
>
> [1] huggingface:falcon-mamba-7b
> [2] huggingface:Jamba-v0.1
>
> Regarding the model’s hidden dimension, we’ve set to 1024, which aligns with the hidden size used in CLIP. Further increasing the hidden dimension would result in a denser Mamba component, making training more challenging, as it would primarily increase parameters related to MLPs.
>
> We also conducted a comparison to assess the impact of increasing Mamba’s capacity by expanding the hidden dimension. With nearly double the number of parameters, the improvement was marginal.
>
> |Hidden Dim|Top-1|Top-5|
> |-|-|-|
> |1024|83.7|96.5|
> |2048|83.9|96.5|
>
> Additionally, we’ve provided a new ablation on SeqMod operation. Please refer to our response to ``R1(5nFH) #2: Further analysis on SeqMod operation``.
>
> >## 3. Adding a VLM task
>
> Yes. See our response to ``R4(17be) #2.1 Equipping MoMa into MLLMs`` for detail.
>
> >## 4. Experimental Designs
> ### 4.1 Fine-tuning results on K400 and SSv2
> MoMa is an adapter method that conducts parameter-efficient fine-tuning (PEFT). Which means, it always fine-tunes a pre-trained image encoder (eg., CLIP) with a small number of trainable parameters. Therefore, all the experiments in Tab.1-3 are finetuning experiments:
> - Tab.1: Fine-tune CLIP on K400
> - Tab.2: Fine-tune CLIP on SSv2
> - Tab.3: Fine-tune Tab.1 checkpoint on COIN and Breakfast
>
> ### 4.2 Long-video understanding experiment
> Sorry for term mis-use. Our MoMa should be clarified as non-e2e method since it conduct PEFT. We will correct this.
>
> For implementation details, both standard and long-video training follows the details in Sec.4. Batch size varies according to the input frame. When input 32 frames, the batch size is 8/gpu.
>
> ### 4.3 Fix misleading table
> Thanks for kind reminder!
>
> We used dividing lines to distinguish between non-PEFT and PEFT methods. All PEFT methods are based on CLIP and can be fairly compared with ours. For non-PEFT methods, there are both large-scale data based methods (eg., ActionCLIP) and those based solely on ImageNet (eg., ViViT).
>
> To avoid misleading, we'll update Tab.1-3 to include pre-training datasets, and we'll take CLIP pre-training dataset into account.
>
> ### 4.4 12-Hour training time
> Although the CLIP parameters do not require training, they still participate in gradient propagation since the adapter layers are interspersed between the CLIP transformer layers. In fact, for video understanding, 12 hours of training time is already quite fast. For comparison, AIM takes over 16 hours for training with the same number of parameters.
>
> >## 5. Literature review
>
> Thank you for the suggestion. Indeed, there are several excellent linear-complexity sequence models besides Mamba, such as S4, TRecViT and RWKV. We will make sure to include them in the related work section.
>
> >## 6. Typos and unclear notions
>
> We apologies for any confusion caused by the typos and unclear notions. Here are some clarifications:
>
> - The $16*16$ windows are expressed in terms of patches.
> - The frame number 16 refers to extracting 16 frames from a video clip for downstream tasks (randomly sampling for training, average sample for inference).
> - $y$ or $y_1,y_2$: We modify the SSM module to output two sequences $(y_1, y_2)$ during SeqMod operation. For the baseline operations, the SSM module only outputs a single sequence $(y)$.

---

> > ### Comment · Reviewer_jWpG · 2025-04-04
> >
> > Thank you for the rebuttal.
> >
> > The question about state size referred to Mamba state size. 16 was used in language. Video is a different domain, hence an investigation on the influence of Mamba state size could be interesting. Or a discussion on the bottleneck that this fairly small state size brings given that the hidden model dimension is fairly large in comparison (1024).
> >
> > I encourage the authors to use carefully the terms fine-tuning vs PEFT. To me, finetuning means updating end-to-end all the parameters of the model, whereas in PEFT only a subset of parameters are updated, the rest are frozen.

---

> > > ### Author Response · Authors · 2025-04-06
> > >
> > > > The question about state size referred to Mamba state size. 16 was used in language. Video is a different domain, hence an investigation on the influence of Mamba state size could be interesting. Or a discussion on the bottleneck that this fairly small state size brings given that the hidden model dimension is fairly large in comparison (1024).
> > >
> > >
> > > Thank you for comments. There are probably some misunderstandings regarding the Mamba state size, which we would like to clarify.
> > >
> > > The state size in Mamba is not a parameter like the hidden dimension which can be arbitrarily large (e.g., 512 or 1024). Instead, it is conceptually closer to **the number of attention heads** in an attention mechanism. As stated in the Mamba2[1] paper (here N refers to the state size):
> > >
> > > ***"… a single channel of S6 produces N inner attention matrices ([1] Eq. 16).”***
> > >
> > > In this context, 16 is already considered a large value for the state size. The original Mamba[2] paper only explores state sizes from 1 to 16 (shown in its Tab. 10). Additionally, **setting the state size to 16 has also become a consensus in vision-based Mamba architectures**. For example, both Vim[3] and MambaVision[4] use 16 as their state size. VMamba[5] architecture optimize it to 1 to increase throughput. For video tasks, the default state size for VideoMamba[6] is also 16.
> > >
> > > Therefore, we would like to clarify that **16 is not a “fairly small” state size and does not present a bottleneck to our model’s performance**.
> > >
> > > Increasing the state size further can actually significantly **decrease the model’s inference speed**. Since the model's complexity is $O(LN^2)$, where $L$ is the sequence length and $N$ is the state size. In typical settings, $N \ll L$, but if we double the state size, the computational complexity would increase by a factor of 4.
> > >
> > > In response to your suggestion, we performed an additional comparison with a doubled state size. The results are conducted on K400 with ViT-B/16 backbone and 8 frames input.
> > >
> > > | State Size | Top-1 | Top-5 |
> > > | ---------- | ----- | ----- |
> > > | 16         | 83.7  | 96.5  |
> > > | 32         | 83.0  | 95.2  |
> > >
> > >
> > > As shown in the table, increasing the state size from 16 to 32 hurts the performance. This aligns with the findings from the original Mamba paper ([2] Tab. 10).
> > >
> > >
> > > [1]: The Hidden Attention of Mamba Models
> > >
> > > [2]: Mamba: Linear-Time Sequence Modeling with Selective State Spaces
> > >
> > > [3]: Vision Mamba: Efficient Visual Representation Learning with Bidirectional State Space Model (ICML 24)
> > >
> > > [4]: MambaVision: A Hybrid Mamba-Transformer Vision Backbone (CVPR 25)
> > >
> > > [5]: VMamba: Visual State Space Model (NIPS 24)
> > >
> > > [6]: VideoMamba: State Space Model for Efficient Video Understanding (ECCV 24)
> > >
> > >
> > > > I encourage the authors to use carefully the terms fine-tuning vs PEFT. To me, finetuning means updating end-to-end all the parameters of the model, whereas in PEFT only a subset of parameters are updated, the rest are frozen.
> > >
> > > Thank you for your suggestion. We will be more careful in differentiating fine-tuning and PEFT and avoid misuse. We have updated our script accordingly to avoid any confusion and ensure greater rigor in our terminology.
> > >
> > >
> > >
> > > Finally, we would like to sincerely thank you for your detailed review and valuable feedback. We truly appreciate the time and effort you invested in carefully reading our paper and providing insightful comments on both our experiments and writing. We hope our response addresses your concerns, and we would be grateful for your consideration in increasing the score.

---

### Official Review · Reviewer_5nFH · 2025-03-14

**Overall Recommendation:** 3

**Summary:**

This paper presents MoMa, a video foundation model that is built on top of the image foundation model by leveraging Mamba as an efficient adapter. Specifically, Mamba block is used to capture spatial-temporal dynamics without interfering with the pre-trained IFMs. Besides, to avoid excessive computational overhead, the authors also use window attention to reduce the attention module’s computation cost. The proposed MoMa shows competitive results on video classification benchmarks.

**Claims And Evidence:**

Yes.

**Essential References Not Discussed:**

I do not identify any important but missing references in this paper.

**Experimental Designs Or Analyses:**

The experimental design in this paper looks reasonable and makes sense.

**Methods And Evaluation Criteria:**

Yes, using a sub-quadratic attention module for spatial-temporal modeling makes sense.

**Other Comments Or Suggestions:**

Please refer to weaknesses.

**Other Strengths And Weaknesses:**

**Weaknesses**

1. The improvement over previous methods appears to be quite marginal on both the K400 and SSv2 datasets.
2. The results in Table 5 do not seem entirely reasonable, as SeqMod shows an improvement of more than 10 points compared to each alternative design, despite their mathematical structures being quite similar. I suggest the authors conduct further analysis and provide a theoretical explanation for this discrepancy.

**Questions For Authors:**

Please refer to weaknesses.

**Relation To Broader Scientific Literature:**

The design of MoMa and SeqMod are both highly related to the efficient video modeling field.

**Theoretical Claims:**

This paper does not provide any theoretical proof.

---

> ### Author Rebuttal · Authors · 2025-04-01
>
> >## 1. Marginal improvement on K400 and SSv2 datasets
>
> We focus on striking a balance between performance and efficiency instead of solely pursuing the highest performance. Besides achieving state-of-the-art performance, we managed to cut nearly 30% calculation FLOPs off with minimum training parameters compared with other PEFT methods. Furthermore, our speed advantage becomes increasingly pronounced with more frames while sustains its performance superiority, as demonstrated in Figure 4.
>
> We also conduct another experiment to demonstrate our model's potential by equipping MoMa with MLLMs and answering complicated video understanding tasks. Please see our response to ``R4(17be) #2.1 Equipping MoMa into MLLMs`` for detail.
>
> >## 2. Further analysis on SeqMod operation.
>
> We apologize for the omission of a comparison with the raw AdaN method and some explanation in our initial submission. Here we supplement more detailed comparative experiments.
>
> | Number | Methods  | Operation| Top-1 | Top-5 |
> | --| -------- | -| ----- | ----- |
> | 1 | Skip     | $x$ | 72.4  | 90.8  |
> | 2 | Add      | $x + y$ | 69.3  | 88.7  |
> | 3 | Max      | $max(x, y)$ | 70.2  | 89.6  |
> | 4 | Concat   | $Linear([x, y])$  | 75.5  | 92.5  |
> | 5 | Raw-AdaN | $\alpha_y\cdot x+x+\beta_y$ | 78.9  | 94.1  |
> | 6 | SeqMod   | $\mathbf{y_1} \odot x + x + \mathbf{y_2}$ | 84.8  | 96.5  |
>
> The Raw-AdaN method learns scalar parameters $\alpha_y$ and $\beta_y$ for adaptation. The table shows that the Raw-AdaN already outperforms other methods like Add, Max, etc. Inspired by its promising results, we further design SeqMod in our paper. We here give some detailed discuss.
>
>
> ### 2.1. Why Raw-AdaN superior Add/Max/Concat, etc.
>
> This is because the feature spaces of Transformer and Mamba do not match or are not perfectly aligned, as observed in [1]. Direct Add/Max/Concat operations interfere with the feature of the original Transformer model.
>
> [1]: ReMamber: Referring Image Segmentation with Mamba Twister
>
> * Add/Max: Adding the output sequence of SSM $y$ to the original features $x$ can lead to information “confusion”, especially when the two features are misaligned or have inconsistent distributions. These two operations can cause information overlap, leading to the loss of key details, especially in multi-scale spatial and temporal features.
>
> * Concat: While it preserves more information by adding learnable linear layer, it does not guarantee that the modification of the sequence is orthogonal.
>
> Different from above, Raw-AdaN uses a global scalar modulation. The scalar acts as a “soft gating” mechanism, where the changes to the sequence’s features are **orthogonal** to the original CLIP feature outputs. This helps avoid altering the feature distribution of the original CLIP outputs when learning global spatiotemporal representations, reducing interference with pre-trained knowledge.
>
> ### 2.2 Why SeqMod significantly outperform Raw-AdaN
>
> **Fine-grained spatiotemporal awareness for video tasks.**
>
> Spatiotemporal modeling is the core to video understanding tasks. Though powerful, Raw-AdaN’s global scalar adjustment (with the same scaling and bias parameters shared across all positions) cannot distinguish dynamic changes in different regions, which is essential for video tasks.
>
> By using fine-grained modulation (retaining independent spatiotemporal modulation parameters at each position), we can capture more complex spatiotemporal dynamics, thus preventing information loss.
>
>
> **Experimental results**
>
> The table above (lines 5 and 6) shows that extending Raw-AdaN’s global scalar modulation to SeqMod’s fine-grained spatiotemporal modulation significantly improves model performance (both $y_1$ and $y_2$ are vectors generated by SSM).
>
> **To sum up:**
>
> 1. AdaN-like methods bring **orthogonal** modulation to the original CLIP, making it more suitable for adaptation tasks and avoid altering the feature distribution too much.
> 2. SeqMod further extends the AdaN-like methods to **fine-grained** spatiotemporal modulation, which is more suitable for video tasks.
>
> We will add the ablation and the according discussion in our final version.

---

### Decision · Program_Chairs · 2025-05-01

**Decision:**

Accept (poster)

**Comment:**

This paper proposes, MoMa, to adapt an Image Foundational Model (such as CLIP, ViT-MAE) for video classification & understanding by integrating Mamba’s selective State Space Modeling into image foundational models. Experiments are done on video classification standard benchmarks for both short and long video classification. During rebuttal, the authors also further provide additional experiments on video QA with MLLMs. After rebuttal, all reviewers recommend to accept the paper. AC reads all reviews, rebuttal and discussions between authors and reviewers. AC agrees with the recommendations from the reviewers, thus recommends to accept this paper.

Note to authors: please incorporate the additional experiments and ablations into the final camera-ready version of the paper if it is accepted.